# Effect of Hypotension and Dobutamine on Gastrointestinal Microcirculations of Healthy, Anesthetized Horses

**DOI:** 10.3390/vetsci11020095

**Published:** 2024-02-19

**Authors:** Philip J. Kieffer, Jarred M. Williams, Molly K. Shepard, Steeve Giguère, Kira L. Epstein

**Affiliations:** 1Evidensia Specialisthästsjukhuset Helsingborg, Bergavägen 3, 254 52 Helsingborg, Sweden; 2Department of Large Animal Medicine, College of Veterinary Medicine, University of Georgia, Athens, GA 30602, USA; jarred@uga.edu (J.M.W.);; 3MedVet Medical & Cancer Centers for Pets, Chicago, IL 60618, USA

**Keywords:** horse, colic, hypotension, perfusion, microvascular, dobutamine

## Abstract

**Simple Summary:**

When horses undergo abdominal surgery, there is a risk that tissues may not receive enough blood. One way to decide if this is happening is to measure blood pressure. The assumption is that if blood pressure is normal, then tissue blood flow is fine. A technique called dark-field microscopy has been used to look at the blood flow to tissues. We measured blood flow at three different places (the gums, the rectum, and the outside of the colon) of six healthy horses undergoing general anesthesia. Additionally, we measured how blood flow to these tissues changed when the patient’s blood pressure was lowered by giving an increased amount of gas anesthesia, which was subsequently raised with a drug called dobutamine. We found that blood flow to these tissues was present when the patients had normal blood pressure. When the blood pressure decreased, the blood flow to the tissues was unaltered. Finally, when blood pressure returned to normal, the blood flow to the tissues remained unaltered. Because blood flow to the tissue was unchanged despite clinically relevant changes in blood pressure, we concluded that using blood pressure as the only method to assess tissue blood flow may be inaccurate.

**Abstract:**

Horses undergoing abdominal exploratory surgery are at risk of hypotension and hypoperfusion. Normal mean arterial pressure is used as a surrogate for adequate tissue perfusion. However, measures of systemic circulation may not be reflective of microcirculation. This study measured the mean arterial pressure, cardiac index, lactate, and four microcirculatory indices in six healthy, anesthetized adult horses undergoing elective laparotomies. The microcirculatory parameters were measured at three different sites along the gastrointestinal tract (oral mucosa, colonic serosa, and rectal mucosa) with dark-field microscopy. All macro- and microcirculatory parameters were obtained when the horses were normotensive, hypotensive, and when normotension returned following treatment with dobutamine. Hypotension was induced with increases in inhaled isoflurane. The horses successfully induced into hypotension did not demonstrate consistent, expected changes in systemic perfusion or microvascular perfusion parameters at any of the three measured gastrointestinal sites. Normotension was successfully restored with the use of dobutamine, while the systemic perfusion and microvascular perfusion parameters remained relatively unchanged. These findings suggest that the use of mean arterial pressure to make clinical decisions regarding perfusion may or may not be accurate.

## 1. Introduction

Horses undergoing abdominal exploratory surgery for colic are at risk of global and local tissue hypoperfusion. In cases with gastrointestinal disease, the hypoperfusion of the gastrointestinal microcirculation and local ischemia can contribute to gastrointestinal dysfunction and tissue death. Changes in cardiovascular function in these horses can be the result of a variety of circumstances, including, but not limited to, anesthesia or endotoxemia [1,2,3]. These changes can lead to generalized poor perfusion throughout the body. In horses with gastrointestinal disease, this issue can be compounded by alterations in the local tissue environment (endothelial dysfunction and inflammation) associated with the primary disease [4].

The early, intra-operative identification of hypoperfusion is necessary to allow intervention and prevent complications. However, traditional monitoring does not directly assess perfusion in general, let alone within the microcirculation. Blood flow or perfusion is the result of a balance between blood pressure and vascular resistance. Systemically, blood flow is equal to cardiac output, but the routine measurement of cardiac output is uncommon due to limitations in the availability and costs of monitoring techniques. In practice, arterial blood pressure can be measured easily and accurately, making it one of the most commonly used techniques for evaluating cardiovascular function [5]. However, hypoperfusion can occur in cases of normotension, and normal perfusion can occur in cases of hypotension depending on the relative change in vascular resistance. Additionally, as noted, perfusion to specific organ beds or even at different locations within a given organ bed may not be predicted using global perfusion/cardiovascular parameters. Thus, the monitoring of local microvascular circulation could improve the identification of abnormalities in tissue perfusion.

Side stream dark field microscopy has been developed for use as a clinical tool to assess microcirculatory perfusion. It has been evaluated in human and veterinary patients, specifically dogs and horses [6,7,8,9]. This device emits a green light (530 nm) that is absorbed by the hemoglobin of erythrocytes while depolarized reflected light from the surrounding tissues is projected back to the device [10]. These illuminated red blood cells can be seen flowing through the reflective microcirculation when viewed through the device, such that red blood cells are of a dark density on a white background [10]. As a result, a real-time video image of red blood cells flowing through the microcirculation with 326× magnification is obtained [10]. These images provide immediate, subjective information about tissue perfusion, and videos can be analyzed to determine several objective microvascular perfusion indices (MPIs), including the total vessel density (TVD), the perfused vessel density (PVD), the proportion of perfused vessels (PPV), and the microvascular flow index (MFI). The sublingual microcirculation has been the most popular region for study in humans and small animals [8,9,11,12,13,14]. In horses, oral and rectal mucosa and colonic serosa have been evaluated in healthy awake and sick anesthetized animals, respectively [6,7]. Microvascular perfusion indices of the oral mucosa, rectal mucosa, and colonic serosa were evaluated concurrently in healthy normotensive anesthetized horses, showing no significant differences between the sites [15].

When hypotension occurs, a variety of treatments are employed to improve blood pressure with the goal of improving perfusion. Such treatments include but are not limited to, intravenous fluid therapy, decreasing the amount of the inhalant administered, and the administration of a cardiovascular stimulant and/or vasoactive drugs. Dobutamine, a beta-1 adrenergic receptor agonist acting on the myocardium in improving contractility, is one of the most commonly used medications to improve blood pressure in horses under general anesthesia [16]. Despite the typical success of dobutamine to improve systemic low blood pressure, hypoperfusion may be ongoing and undiagnosed. An evaluation of the microcirculation with dark field microscopy has the potential to more accurately describe the effects of systemic hypotension and dobutamine treatment on tissue perfusion. Therefore, the objectives of this study were to evaluate (1) the effects of systemic hypotension on dark field microscopy MPIs in three microvascular beds and (2) the effects of the correction of hypotension with dobutamine on dark field microscopy MPIs. Our hypotheses were that in healthy horses undergoing general anesthesia, hypotension would decrease the MPIs within each site, and the correction of hypotension with dobutamine would return the MPIs to values recorded during normotension without cardiovascular support drugs.

## 2. Materials and Methods

### 2.1. Animals, Anesthesia and Monitoring

This study was approved by the University of Georgia Institutional Animal Care and Use Committee (Approval Code: A2014 05-022-Y3-A0). Six healthy adult (average age 17; range 10–29 years) horses (3 mares and 3 geldings) without known gastrointestinal disease or cardiovascular compromise were included in this study (average weight 468 kg; range 418–575 kg). Each horse underwent a physical examination and was deemed systemically healthy before inclusion in the study. Horses had an intravenous catheter placed in the right jugular vein and were pre-medicated with intravenous xylazine hydrochloride (1.1 mg/kg); anesthesia was induced with intravenous diazepam (0.05 mg/kg) and ketamine hydrochloride (2.2 mg/kg). Once anesthetized, the horses were intubated orotracheally and placed into dorsal recumbency. Isoflurane was delivered in 100% oxygen, and mechanical ventilation was used to maintain end-tidal CO_2_ between 35 and 45 mmHg. Intravenous polyionic fluids were delivered initially at 10 mL/kg/h. The direct cannulation of the facial artery for direct blood pressure monitoring was performed and maintained throughout the study period. In addition to blood pressure, vital parameters, electrocardiography, end-tidal CO_2_, oxygen flow rate, inspired oxygen concentration, tidal volume, exhaled isoflurane concentration, and peak inspiratory pressure were recorded every 5 min. Electrolyte concentration, blood gas analysis, and cardiac output (taken in duplicate via lithium dilution) [17,18] were obtained at the time of collection of the microcirculatory variables. Cardiac output was measured in duplicate via lithium dilution with a LiDCO computer (LiDCO plus, cardiac computer, LiDCO Group PLC, London, UK) [17]. The cardiac index was calculated by dividing the cardiac output by the weight (kg) of the horse. The adjustment of the rate of fluid delivery and percentage of inhaled isoflurane at the discretion of the anesthesiologist was used to maintain normotension (MAP 70–90 mmHg) for the first time point. Hypotension (MAP < 60 mmHg) was then induced by increasing the administered isoflurane for the second time point. Dobutamine was administered via a continuous rate infusion at a rate (0.5 mcg/kg/min) that was titrated to effect until normotension (MAP 70–90 mmHg) was achieved. Measurements were repeated once, and normotension was achieved with the third and final time point.

### 2.2. Dark Field Microscopy Image Acquisition

Microcirculation videos (Microscan; MicroVision Medical, Amsterdam, The Netherlands) were obtained from the oral mucosa, rectal mucosa, and the serosa of the pelvic flexure region of the large colon. Videos were collected according to a previously described protocol [15,19]. The videos were obtained when the patients were in a normotensive state without the administration of any cardiovascular support drugs (normotension), induced hypotension (hypotension), and dobutamine-induced normotension (dobutamine). For all measurement periods, the respective blood pressures were maintained for at least 10 min prior to and then throughout image collection.

To access the pelvic flexure, a ventral midline celiotomy was performed. The abdomen was clipped and aseptically prepared, and a sterile drape was placed. A 20 cm long ventral midline incision was made with a scalpel. The pelvic flexure was exteriorized onto an enterotomy (colon) tray to allow for image collection. The pelvic flexure was returned to the abdomen between image collection time points. The tray was positioned downward at a gentle angle from the incision that was considered clinically appropriate to the horse’s body size and conformation, allowing the colon to rest without tension. The angle was subjectively based on the clinical experience of PK, KE, and JW.

For videos obtained from the colon, the colon was placed on the enterotomy tray, and the tip of the probe was gently placed against the colonic serosa at the pelvic flexure. The probe was stabilized for collection using sandbags to support the unit. Once the unit was placed, the operator was able to remove their hands from the unit during collection in most cases.

For videos obtained from the rectum, the probe was manipulated and inserted at least half the distance of the disposable protective cover (approximately 1–2 cm in length). Gross fecal material was digitally removed from the rectum if necessary and lavaged away. The operator then utilized the table as a reference to angulate the probe on the mucosa as ideally as possible for image collection. For videos obtained from the oral mucosa, the probe was placed on the gingival mucosa and stabilized using sandbags. For all locations, warmed sterile isotonic fluids (0.9% NaCl) were applied as needed to the tissue and abdomen at the discretion of the investigators to maintain tissue integrity and image quality. Video loops were collected around the mechanical ventilation of the horse to minimize motion.

A minimum of 3 videos, 20 s in length, were taken from each location at each sample time. The videos were obtained by the same unit. The videos for each site were obtained sequentially, and the order of site acquisition was randomized. The probe was moved between each video to adjacent but similar sites for subsequent image collection. Following the completion of data collection for this study, additional data/samples were collected from the subjects for other approved projects, and subjects were then humanely euthanized while under general anesthesia.

### 2.3. Measurement of Microvascular Perfusion Indices

Analysis was performed in accordance with the De Backer et al. roundtable [19]. Three videos of at least 50 frames were produced from the full videos for each site. Video selection was based on clarity, stability, and acceptable image quality. The videos were then blinded by one investigator (JMW). An analysis of the blinded videos was performed by one investigator (PJK) to determine microcirculatory variables (as previously described based on consensus criteria [8,19]) using manufacturer-provided software (Automated Vascular Analysis, Version 3.2; MicroVision Medical, Amsterdam, The Netherland). Briefly, this included the total vessel density (TVD), the proportion of perfused vessels (PPV), perfused vessel density (PVD), and microvascular flow index (MFI). Detailed descriptions of video analysis and how these values are determined can be found in numerous sources [7,8,19].

### 2.4. Data Analysis

The normality of data was assessed based on an examination of histograms, the normal Q–Q plots of the residuals, and the Shapiro–Wilk test. The variance of the data was assessed by plotting residuals against predicted values and using Levene’s test. The effect of the blood pressure state (normotensive, hypotensive, dobutamine) on MAP, CI, lactate, and microvascular variables was assessed using linear marginal models with compound symmetry, heterogeneous compound symmetry, or unstructured covariance structures. To assess the repeatability and measurement of microvascular variables, the coefficient of variation in triplicate measurements was calculated for each horse, site (colon, oral, or rectal), and microvascular variable. The effects of the site, microvascular variable, and interaction between site and microvascular variable on the CV were evaluated using a linear mixed-effects model with the horse modeled as a random effect and the site and microvascular variable modeled as fixed nominal effects. When indicated, multiple pairwise comparisons were performed using the method by Sidak. The association between colonic, oral, and rectal microvascular variables and the association between microvascular variables and MAP or CI were investigated by calculating correlation coefficients using the method described by Bland and Altman to account for repeated observations from individual animals [20]. For all analyses, *p* < 0.05 was considered significant. All statistical analyses were performed using SPSS (SPSS version 23. IBM Corp., Armonk, NY, USA).

## 3. Results

The duration of time between the induction of anesthesia to the start of video collection was an average of 37 min. The duration of time required for the collection of all the videos averaged 102 min (range 84–125 min). This time included device movement between sites, the exteriorization of the colon prior to obtaining videos, and the subsequent return of the colon to the abdomen following video collection. While the specific time to record the video triplicates at each site during each collection time was not measured, subjectively, these three videos were obtained within five minutes each time. The duration of time that the dobutamine was administered averaged 27.17 min (range 20–35 min). The average rate of dobutamine administered was 0.76 mcg/kg/min (0.27–2 mcg/kg/min). Efforts were made to maintain a consistent end-tidal isoflurane concentration; however, the isoflurane dose was modified to some degree during data collection in order to ensure that the blood pressure and the amount of dobutamine administered remained within the desired range. The intravenous fluid rate average was 3.7 L per hour (2–10 L/h), with an average total volume of 9.5 L per horse (7–17 L/horse). The average hematocrit (HCT) was 31.2% (26–34%), with an average change in HCT of 3.3% (1–6%).

All data for mean arterial pressure, cardiac index, and lactate were normally distributed. Significant differences existed in the MAP between normotensive (mean: 81 +/− 2 mmHg) and hypotensive (mean: 49 +/− 2 mmHg) (*p* < 0.000001) time points and between the dobutamine (mean: 81 +/− 1 mmHg) and hypotensive time points (*p* < 0.000001). No significant difference existed between the normotensive and dobutamine time points. No significant difference in the cardiac index was present between the normotensive (mean: 101.39 mL/kg/min) and hypotensive (mean: 94.26 mL/kg/min) time points, but a significant difference was detected between the dobutamine (mean: 115.79 mL/kg/min) time points and both the normotensive and hypotensive time points, *p* = 0.017 and *p* = 0.0016, respectively. The mean lactate concentrations for the normotensive, hypotensive, and dobutamine time points were 0.97, 1.08, and 1.25 mmol/L, respectively. A statistically significant difference (*p* = 0.017) was present in lactate levels between the dobutamine and normotensive time points, although levels remained within the clinically normal reference range (<2 mmol/L). No significant differences were present in the lactate levels between dobutamine and hypotensive time points or between hypotensive and normotensive time points.

All data for the microvascular perfusion indices were normally distributed. The mean values for the MPIs (with standard error) are provided in Table 1. No significant associations between TVD, PVD, or MFI were present between blood pressure states in the colon serosa, oral mucosa, and rectal mucosa (Figure 1, Figure 2 and Figure 3, respectively). Significant associations were present between the proportion of perfused vessels of the oral mucosa at the hypotensive and dobutamine time points (*p* = 0.045) but not between other pressure states at this site (Figure 4). Additionally, no significant associations in PPV were present between the pressure states in the colon serosa and rectal mucosa (Figure 4). There were no correlations for any MPI between the oral mucosa, colon serosa, and rectal mucosa. Oral TVD and colon PVD were significantly (*p* < 0.05) correlated with the cardiac index (0.503 and 0.407, respectively). No other MPI at any site correlated with cardiac index, and no microvascular perfusion indices at any site correlated with mean arterial pressure. Complete correlation data are presented in Table 2.

## 4. Discussion

We found that hypotension did not decrease microvascular perfusion indices at any gastrointestinal location, thereby disproving our first hypothesis and making it impossible to test the second hypothesis in which dobutamine would return the MPIs to values similar to those recorded during normotension. In anesthetized, healthy horses, there was minimal to no correlation between macrovascular parameters and microvascular parameters at three gastrointestinal sites during normotension, hypotension, or dobutamine-induced normotension. There were also no correlations between the microvascular parameters between the three gastrointestinal sites evaluated.

In this study, neither the cardiac index nor the microvascular blood flow was altered during periods of systemic hypotension. The lack of change in microperfusion to the regions evaluated was most likely a result of consistent cardiac output, adequate macrovascular flow, and maintained tissue perfusion. Organ tissue perfusion is influenced by the difference between arterial and venous pressure across the organs, and this pressure can vary between organs and peripheral tissues throughout the body. In this study, two peripheral sites (oral and rectal mucosa) and one organ site (colon) were evaluated, and capillary flow was unaltered throughout the varying pressure states. However, it is important to recognize that vascular beds to other organ and peripheral tissue sites may have undergone vasoconstriction in order to maintain cardiac output and adequate blood flow to the sites evaluated.

The lack of correlation between sites is similar to previous work [15]. These results should be interpreted with respect to a small sample size. However, other reasons could also account for the results. The three sites investigated, while all part of the gastrointestinal tract, vary in their access and anatomy. The collection of data with the microscopy unit involved a subjective amount of pressure to be applied to an area. The oral mucosa, with underlying bone, could create greater compression of the microvasculature than areas with softer underlying tissue, such as the colon and rectum. Additionally, the rectal mucosa has more folds and dimensions than the oral mucosa or colonic serosa. These anatomic variations can create scenarios in which variations in image capture may be inconsistent, regardless of their flow to the sites. The microcirculation changes dynamically and continually. Changes in blood flow to one site do not imply identical changes to another. The lack of correlation observed in this study could merely reflect this occurrence. It is important for regional and local vasoconstriction and vasodilation to occur in order to accommodate regional and local demands and to allow for rapid and significant changes in systemic vascular resistance; this is a key feature when compensating for shock.

In this study, hypotension was effectively induced by increasing the percentage of inhaled isoflurane [21]. Isoflurane inhalation alters cardiovascular function via vasodilation and, to a lesser extent, decreases contractility [21]. Vasodilation affects the arterial and venous side of the circulation, but hypotension, which is related to isoflurane anesthesia, is primarily the result of arterial dilation. The overall impact of the vascular tone on cardiac output is difficult to determine; however, due to the contrasting effects of arterial and venous dilation on stroke volume, the decrease in contractility caused by isoflurane is relatively minor and may also have been minimized by compensatory increases in contractility as a physiologic response to the increasing stroke volume.

The use of intravenous fluids to support the preload may have decreased the negative effect of venous dilation on cardiac output. Though intravenous fluids can also have a dilutional effect on hemoglobin, despite the use of intravenous fluids in this study, the hematocrit remained within the normal limits for all horses at all time points and only changed three percent, on average, throughout the study; the biggest change was 6% in one horse. Therefore, despite the potential dilutional effect of intravenous fluids on hemoglobin and perfusion, we do not believe it was a factor in the results of this study.

The isoflurane model for hypotension was chosen due to its reliability, ease of administration, and clinical relevance. Isoflurane is a commonly used inhalant anesthetic for equine general anesthesia, and hypotension is considered to be a common, clinically significant side effect of its use. Despite achieving mean arterial pressures that fulfill the definition of hypotension in our test subjects, systemic lactate, cardiac index, and microcirculatory perfusion indices remained clinically and, with the exception of lactate, statistically unchanged. These findings show that normal, healthy equine patients undergoing elective procedures in dorsal recumbency are likely able to maintain adequate local perfusion of the gastrointestinal tract despite the degree of systemic hypotension that was induced in this study. One limitation of this study is the health status of our test subjects; compared to horses with gastrointestinal disease, clinically healthy horses may respond much differently to isoflurane anesthesia and maintain many different relationships between their systemic cardiovascular and microvascular status. For this reason, repeating this study with different models of induced hypotension, such as blood loss or endotoxin, on clinically ill patients is necessary to determine the effects of hypotension on microcirculatory perfusion indices.

Dobutamine effectively reversed hypotension in this study. Dobutamine is a positive inotrope that increases cardiac output and stroke volume, thereby reversing hypotension. It effectively reversed hypotension in our study population and significantly increased cardiac index. Although the cardiac index was increased, microvascular perfusion indices and lactate values were not significantly altered at the dobutamine time point. This lack of change in microvascular perfusion indices and lactate might be due to a continuation of adequate systemic perfusion. Alternatively, the local control of microvascular flow may have prevented these indices from increasing beyond normal.

One study evaluated the effects of dobutamine at three doses (0.5, 1, and 3 mcg/kg/min) on systemic hemodynamics and intestinal perfusion (jejunum and colon) in healthy horses undergoing anesthesia with isoflurane. Compared to the baseline, the higher doses of dobutamine significantly increased CO, HR, MAP, and blood flow to the jejunum and colon, though increased blood flow to the colon and jejunum was not significantly increased at 0.5 mcg/kg/min of dobutamine [22]. Our results differ from those observed at higher doses in this study, though these are similar to those observed at the low dose. In our study, the average dose of dobutamine administered was 0.76 mcg/kg/min, with the titration of dobutamine creating dose ranges from 0.27 mcg/kg/min to 2 mcg/kg/min. We suspect that the similarities observed between the two studies have much in common with the dobutamine dosage. Additionally, the two studies evaluated the microcirculation differently. As microvascular beds are difficult to assess, a gold standard method has not been determined. We evaluated blood to the colon with side stream dark field microscopy, while others evaluated it with a micro-lightguide spectrophotometer [22]. It is very reasonable to suspect that differences in techniques could account for some difference in results, at least until the two techniques are evaluated against each other.

It is worth mentioning that the effects of other medications given on vascular tone are known and unavoidable [23,24,25]. For this reason, the anesthetic protocol was standardized to minimize any effects of various injectable drugs on the microcirculation. Additionally, as noted, although efforts were made to minimize adjustments to the percentage of isoflurane, it was sometimes required to maintain consistent MAP and/or dobutamine infusions. It is possible that these adjustments could have affected some measurements.

The microcirculatory function is critical to tissue perfusion, but the ability to assess the vascular beds of interest is challenging. Systemic parameters (MAP, CO, HR) can be more easily obtained in clinical patients. Thus, using systemic parameters to accurately assess the microcirculation would be a convenient circumstance. This study, like many studies before it, did not find a relationship between macro and microcirculation; this was especially true for disease states [7,8,26,27,28,29,30,31]. The lack of observable correlation between the systemic variable parameters and microvascular perfusion indices reported in the current study may be due to the use of apparently healthy horses. These patients likely maintained their local autoregulation. An evaluation of these same tissue beds in disease states is interesting and may yield different results.

Treatments such as dobutamine and intravascular fluids have the potential to alter blood volume and hematocrit. If blood viscosity is increased with hemoconcentration or decreased with hemodilution, changes in perfusion can be affected. In this study, the hematocrit was within normal limits throughout. As such, the results of this study are unlikely to be related to alterations in viscosity.

There are several limitations to this study. The use of side stream dark field is limited to mucosal or serosal surfaces, negatively affecting its utility elsewhere, such as the skin or within organs. Peristalsis within the colon and patient ventilation made image capture difficult when accounting for motion artifacts. For this reason, we used the recommended 20 s for assessment but spent a considerable amount of time in image analysis to identify at least 50 continuous frames of quality imaging for analysis. Additionally, the delay in image analysis makes this technology difficult to evaluate in a real-time clinical setting. Finally, a larger sample size could have demonstrated different results. A sample size of six was chosen based on financial and ethical concerns.

## 5. Conclusions

In summary, in healthy, anesthetized adult horses, changes in blood pressure did not result in consistent expected changes in systemic perfusion or microvascular perfusion parameters at the three sites. Based on these results, caution is recommended when interpreting the mean arterial pressure as a reflection of tissue perfusion. However, the most conservative and safe clinical approach while treating equine patients may be to assume decreased microvascular perfusion in the event of systemic hypotension. Additionally, the use of dobutamine was justified, based on the observable impact of that treatment on CI and MAP, without producing negative impacts on microperfusion. The lack of associations between the microvascular sites suggests that the use of one microvascular system to predict the behavior or condition of another microvascular system may lead to inaccuracies.

## Figures and Tables

**Figure 1 vetsci-11-00095-f001:**
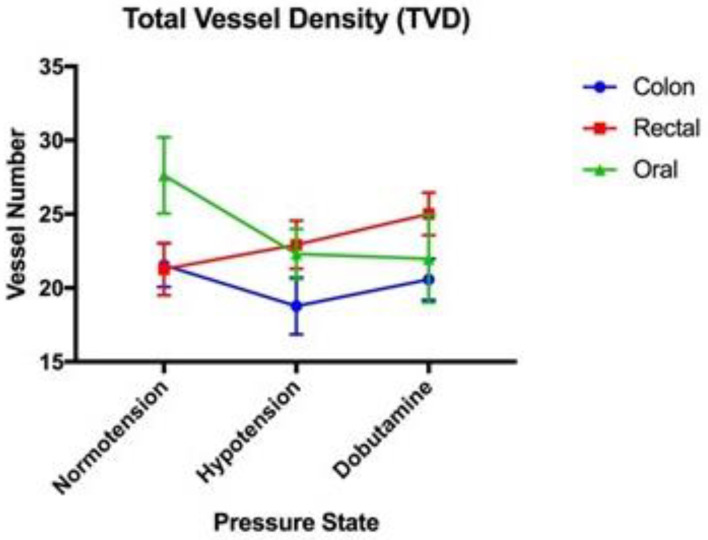
Total vessel density (TVD) (mm/mm^2^): A comparison of the mean and standard error (error bars) of the total vessel density of the oral mucosa, rectal mucosa and colonic serosa under normotension, hypotension, and dobutamine-induced normotension.

**Figure 2 vetsci-11-00095-f002:**
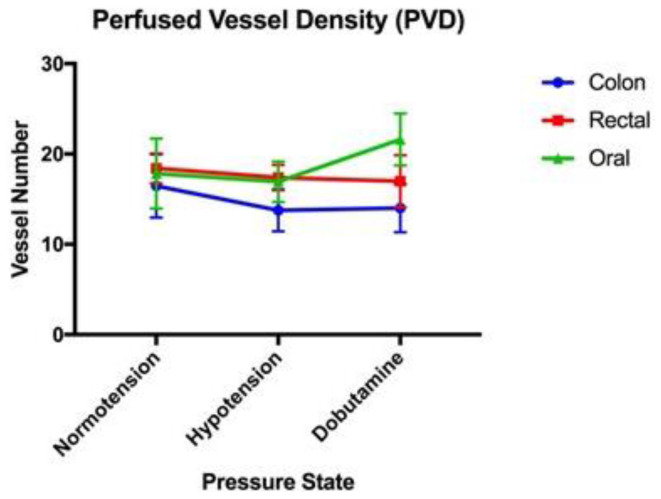
Perfused vessel density (PVD) (mm/mm^2^): A comparison of the mean and standard error (error bars) of the perfused vessel density of the oral mucosa, rectal mucosa, and colonic serosa under normotension, hypotension, and dobutamine-induced normotension.

**Figure 3 vetsci-11-00095-f003:**
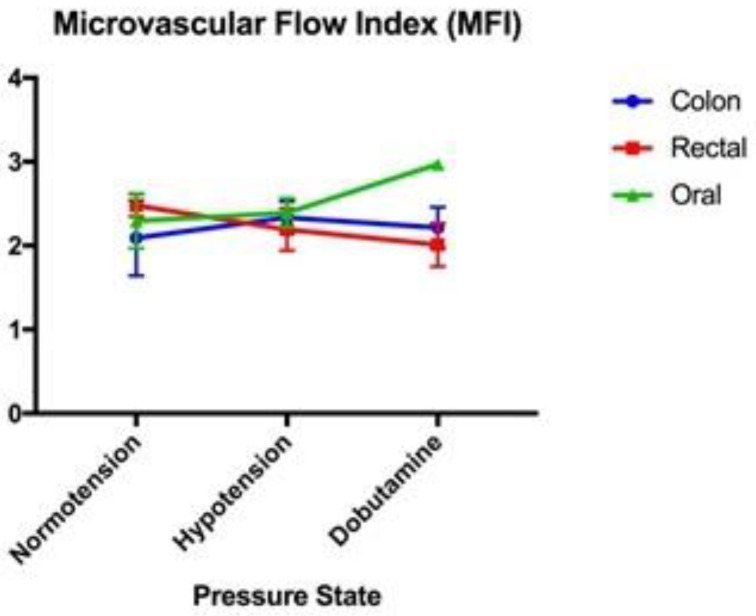
Microvascular flow index (MFI): a comparison of the mean and standard error (error bars) of the microvascular flow index of the oral mucosa, rectal mucosa, and colonic serosa under normotension, hypotension, and dobutamine-induced normotension.

**Figure 4 vetsci-11-00095-f004:**
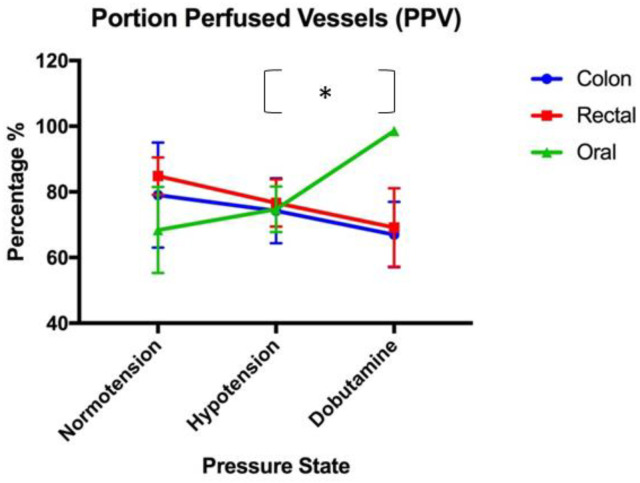
The proportion of perfused vessels (PPV) (%): A comparison of the mean and standard error (error bars) of the proportion of perfused vessels of the oral mucosa, rectal mucosa, and colonic serosa under normotension, hypotension, and dobutamine-induced normotension. The (*) denotes statistically significant differences in the PPV of oral mucosa between hypotension and dobutamine-induced normotension.

**Table 1 vetsci-11-00095-t001:** Microvascular perfusion indices data: The mean (+/−standard error) values for total vessel density (TVD), perfused vessel density (PVD), the proportion of perfused vessels (PPV), and microvascular flow index (MFI)at each site (oral mucosa, colon serosa, and rectal mucosa) during each condition (normotension, hypotension, and dobutamine-induced normotension).

	Normotension	Hypotension	Dobutamine
Oral Mucosa			
TVD (mm/mm^2^)	27.63 (+/−2.59)	22.30 (+/−1.70)	21.99 (+/−2.98)
PVD (mm/mm^2^)	17.84 (+/−3.88)	16.93 (+/−2.24)	21.61 (+/−2.87)
PPV (%)	68.38 (+/−13.10)	74.70 (+/−6.95)	98.52 (+/−0.97)
MFI (units)	2.30 (+/−0.33)	2.39 (+/−0.17)	2.97 (+/−0.07)
Colon Serosa			
TVD (mm/mm^2^)	21.55 (+/−1.48)	18.79 (+/−1.93)	20.58 (+/−1.39)
PVD (mm/mm^2^)	16.47 (+/−3.50)	13.76 (+/−2.32)	14.00 (+/−2.67)
PPV (%)	79.03 (+/−15.98)	74.27 (+/−9.92)	67.00 (+/−9.96)
MFI (units)	2.09 (+/−0.45)	2.33 (+/−0.19)	2.21 (+/−0.25)
Rectal Mucosa			
TVD (mm/mm^2^)	21.28 (+/−1.78)	22.94 (+/−1.63)	25.02 (+/−1.45)
PVD (mm/mm^2^)	18.42 (+/−1.65)	17.38 (+/−1.40)	17.00 (+/−2.88)
PPV (%)	84.80 (+/−5.73)	76.64 (+/−7.21)	69.17 (+/−11.97)
MFI (units)	2.48 (+/−0.13)	2.19 (+/−0.25)	2.01 (+/−0.26)

**Table 2 vetsci-11-00095-t002:** Correlation data: correlations between microvascular parameters (total vessel density (TVD), the perfused vessel density (PVD), proportion of perfused vessels (PPV), and microvascular flow index (MFI)) of different sites (oral mucosa, rectal mucosa, and colon serosa) and between microvascular parameters, the MAP and cardiac index (CI).

Comparison	Correlation Coefficient	*p* Value
Colon TVD vs. Oral TVD	0.177	0.223
Colon PVD vs. Oral PVD	0.209	0.150
Colon PPV vs. Oral PPV	−0.438	0.473
Colon MFI vs. Oral MFI	−0.251	0.082
Colon TVD vs. Rectal TVD	0.114	0.436
Colon PVD vs. Rectal PVD	0.131	0.370
Colon PPV vs. Rectal PPV	0.007	0.961
Colon MFI vs. Rectal MFI	−0.179	0.220
Colon TVD vs. MAP	0.361	0.225
Colon PVD vs. MAP	−0.090	0.770
Colon PPV vs. MAP	−0.133	0.666
Colon MFI vs. MAP	−0.222	0.404
Colon TVD vs. CI	0.002	0.884
Colon PVD vs. CI	0.407	0.019
Colon PPV vs. CI	0.195	0.131
Colon MFI vs. CI	0.013	0.712
Oral TVD vs. MAP	0.033	0.553
Oral PVD vs. MAP	0.077	0.358
Oral PPV vs. MAP	0.053	0.449
Oral MFI vs. MAP	0.069	0.386
Oral TVD vs. CI	0.503	0.007
Oral PVD vs. CI	0.067	0.393
Oral PPV vs. CI	0.248	0.083
Oral MFI vs. CI	0.197	0.129
Rectal TVD vs. MAP	0.004	0.839
Colon PVD vs. MAP	0.036	0.535
Rectal PPV vs. MAP	−0.005	0.814
Rectal MFI vs. MAP	0.003	0.866
Rectal TVD vs. CI	−0.016	0.68
Rectal PVD vs. CI	−0.001	0.938
Rectal PPV vs. CI	0.000	0.948
Rectal MFI vs. CI	0.013	0.71

## Data Availability

The original contributions presented in the study are included in the article, further inquiries can be directed to the corresponding author/s.

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
