# Peer review of "Effect of Hypotension and Dobutamine on Gastrointestinal Microcirculations of Healthy, Anesthetized Horses"

_vetsci, 2024, doi:10.3390/vetsci11020095_

Round 1
Reviewer 1 Report
Comments and Suggestions for Authors
Thank you for presenting a paper which attempts to address our limitations of assessing effects of drugs on microvascular perfusion. I feel the reader would benefit from some additional information in introduction, methods, results and discussion to fully understand the outcome of this study. these are outlined in the comments below
1. Line 60-69: the microvasculature consists of arterioles, capillaries and venules…can this technique distinguish between these structures and if not how do you determine the difference between change in arteriole, capillary and venule blood flow during drug administration;
2. some microvasular beds have AVA that can be opened during drug administration (isoflurane, dobutamine) does this occur in GIT and if so could this affect the indices measured
3. Line 70 dobutamine is known to increase PCV. Indicate how this will affect the indices measured
4. Line 141 As the concentration of sodium and chloride are higher than horse, clarify if the effect of 0.9 % saline on microvascular indices has been determined.
5. line 145 Clarify whether this was 3 videos from each site at each sample time
6. 152-158: clarify which indices were measured, and if 3 videos were collected at each site at each sample time, were the indices averaged over the 3 videos; clarify what information is extracted from each of the 50 frames during analysis and how it is used to calculate the indices
7. Line 173-180; please also indicate the time to collect videos at each site at each sample time; this is particular during dobutamine administration as temporal effects can occur; clarify if the MAP was calculated from an average of readings over this time and if so the MAP was stable ( ie within 10 %)
8. Line 173-180 duration and dose of dobutamine actually given including range should also be provided to demonstrate if any temporal effect was possible
9. Also include fluid rates provided and total dose. Is it possible this had any effect on Hb through dilution.
10. 182-191 some measure of variance should be included for all mean either standard deviation or 95 % confidence interval; the number of decimal places for CV variables are not consistent with standard formatting.
11. Line 192 the p value should be given
12. Blood gas analysis was performed, so measures should include changes in Hb concentration…I would recommend these be included at each of the sample time to demonstrate if PCV could have had an effect on perfusion
13. Line 243-248. Organ tissue perfusion is influenced by difference between arterial and venous pressure across the organ and can be different between organs. Thus, the presence of lack of change in CO may not be relevant, particularly if certain vascular beds are constricted to maintain return of blood volume to heart. Please consider this in your discussion.
14. Line 256 I do not agree with this statement. Vasodilation may decrease afterload and improve cardiac output if it offsets increases in afterload. However in this study the arterial pressure was well below normal and thus organ perfusion pressure would be reduced unless it is able to autoregulate in the presence of isoflurane.
15. Limitations of the technique for measuring perfusion has not been discussed and needs to be included.
a. Can it distinguish between arterial and venous side of circulation and the differences in effect of drugs on each;
b. what is the impact of changes in PCV.
i. Dobutamine can increase PCV. This has the potential for 2 effects that are worthy of consideration i) can this make vessels more obvious ii) could increase viscosity which could offset any potential increases in perfusion associated with MAP. This second effect was first suggested by Colin Dunlop in his dobutamine studies and have been mentioned in several studies since
ii. Hemodilution is known to increase microvascular perfusion by decreasing viscosity, so the effect of fluid volumes on these results need to be addressed.
16. It is recommended that the authors provide more information on the method used to measure perfusion including repeatability, usefulness in face of vessel identification, and changes in PCV on perfison measurements and limitations of the technique to justify the need to perform a non-recovery study
Comments on the Quality of English Language
There were some very minor typographical errors
The authors do need to look at the recommended standards for presenting data ie mean + some indication of variance for all variables (MAP, CI etc) and the number of decimal places considered appropriate.
Reviewer 2 Report
Comments and Suggestions for Authors
The manuscript is well designed and written. The discussion clearly explained the results observed from the study. From the result of the study by Dancker, C et al, (Effects of dobutamine, dopamine, phenylephrine and noradrenaline on systemic haemodynamics and intestinal perfusion in isoflurane anaesthetized horses. Equine Vet Journal 2018;50:104-110), demonstrated that dobutamine infusion increased the blood flow to jejunum and colon, though the increase were more significant at higher doses of dobutamine (1 and 3 ug/kg/min) but not at lower dose (0.5 ug/kg/min), a dose similar to that used in this manuscript. Therefore, my only suggestion for the authors is to include the comparison of the result of the dobutamine infusion in the discussion in this manuscript.
Reviewer 3 Report
Comments and Suggestions for Authors
The authors present an interesting study evaluating the effects of GA and hypotension on microvascular perfusion indices. I am a little confused by the reference to horses with colic, as (as you point out) these were healthy horses. I would suggest removing the discussion of horses with colic and instead reframe this as an important further step in evaluating the effects of techniques/drugs/positioning/disease states on microvascular perfusion. I also think that the conclusion is somewhat exaggerated and perhaps the authors should stick a bit more to what the data demonstrates - that isoflurane induced hypotension, in the absence of changes in cardiac output, does not appear to significantly affect MPIs. It would be interesting to alter cardiac output to decrease blood pressure (instead of vascular tone) and see if the same lack of relationship holds.
Some other specific comments:
line 15 - the way this line is worded makes it sound like the horses were undergoing abdominal surgery for medical reasons - not having their colons exteriorized for purposes of the study prior to euthanasia - please rephrase
line 35 - can you reference this statement about prioritizing blood flow please, especially given that these are heathy horses undergoing GA, not shocky horses or ones with GI disease.
lines 55-60 - can you describe a bit better the technique of dark field microscopy here? I'm not sure how you went from 5x to 326x magnification or how absorption of green light leads to illumination of the RBCs?
line 75 - medications (not medication)
line 89 - please include the IACUC protocol # here for reference. How were these horses chosen for inclusion?
Was a catheter placed prior to premedication? Or at all?
lines 100-105 - I believe these should be separated by commas, not semicolons. Is 'collected' correct? Perhaps recorded? Can you briefly describe the lithium dilution technique and specifics - I am assuming you used a LIdCO plus? volume of lithium? The reference here (17) doesn't really make sense - aren't you trying to reference that lithium dilution accurately measures CO as well as referencing the technique used? Perhaps this study is a better reference: Linton RA, Young LE, Marlin DJ, Blissitt KJ, Brearley JC, Jonas MM, O'Brien TK, Linton NW, Band DM, Hollingworth C, Jones RS. Cardiac output measured by lithium dilution, thermodilution, and transesophageal Doppler echocardiography in anesthetized horses. Am J Vet Res. 2000 Jul;61(7):731-7. doi: 10.2460/ajvr.2000.61.731. PMID: 10895891.
line 108 - I am assuming this 70-90mmHg is MAP like the others?
line 110 - if dobutamine was administered at 0.5mcg/kg/min, how was it titrated? Or was it started at that dose and titrated up or down as necessary? Please also add the dobutamine dose(s) to the results section.
I am assuming that your linear models incorporated the horse as a random effect to account for multiple measurements on each?
line 169 - can you add the B&A reference to the reference list
What version of SPSS was used?
In the results, Figure 1 & 2 don't add anything and the numbers are in the text, I would remove those 2.
The numbers reported for CI are actually CO - unless your horses had a stroke volume of 660 liters??
When discussion the correlations, perhaps you could include a table of the results? Using p-values as a strict cutoff perhaps loses some of the value in the data.
The legend for Table 1 should include definitions for the abbreviations used (TVD, PVD, etc)
Figure 3 shows up very small - perhaps just an editing issue - and needs a legend for the colors. Also perhaps add a bracket under the star to denote that it is referencing two measurements and not just the space in-between.
Line 241 - You note that there was no correlation found between the 3 sites for the MPIs - and then don't discuss this important finding. Please add this to the discussion, and perhaps remove the definition of cardiac output and stroke volume (lines 246-249)
Comments on the Quality of English Language
Manuscript is generally well written with the exceptions noted in the comments to the authors.
Round 2
Reviewer 1 Report
Comments and Suggestions for Authors
I am happy for the paper to be accepted for publication